# OpenDataVal: a Unified Benchmark for Data Valuation

**Kevin Fu Jiang**[*]
Columbia University

**Weixin Liang**[*]
Stanford University

**James Zou**
Stanford University

**Yongchan Kwon**[†]
Columbia University

## Abstract

Assessing the quality and impact of individual data points is critical for improving model performance and mitigating undesirable biases within the training dataset. Several data valuation algorithms have been proposed to quantify data quality, however, there lacks a systemic and standardized benchmarking system for data valuation. In this paper, we introduce `OpenDataVal`, an easy-to-use and unified benchmark framework that empowers researchers and practitioners to apply and compare various data valuation algorithms. `OpenDataVal` provides an integrated environment that includes (i) a diverse collection of image, natural language, and tabular datasets, (ii) implementations of eleven different state-of-the-art data valuation algorithms, and (iii) a prediction model API that can import any models in scikit-learn. Furthermore, we propose four downstream machine learning tasks for evaluating the quality of data values. We perform benchmarking analysis using `OpenDataVal`, quantifying and comparing the efficacy of state-of-the-art data valuation approaches. We find that no single algorithm performs uniformly best across all tasks, and an appropriate algorithm should be employed for a user's downstream task. `OpenDataVal` is publicly available at `https://opendataval.github.io` with comprehensive documentation. Furthermore, we provide a leaderboard where researchers can evaluate the effectiveness of their own data valuation algorithms.

## 1 Introduction

Real-world data often exhibit heterogeneity and noise as they are collected from a wide range of sources and are susceptible to measurement and annotation errors [25, 19, 21]. When such data are incorporated into model development, they can introduce undesirable biases and potentially hinder model performance. Managing and addressing these challenges is crucial to ensure the reliability and accuracy of the insights derived from real-world data. As such, it is becoming increasingly important to understand and evaluate the intrinsic properties of data, such as data quality, data bias, and their influences on the model training process [14, 20]. From all these motivations, data valuation has emerged as a principled evaluation framework that aims to quantify the impact of individual data points on model predictions or model performance.

There have been substantial recent efforts in developing different data valuation methods, and they have demonstrated promising outcomes in various real-world applications. One widely used data valuation method is DataShapley [7], which is based on the Shapley value from game theory [31]. This method has been applied to detect low-quality chest X-ray image data [35] and design a data

---

[*]Co-first author

[†]Corresponding author (e-mail: yk3012@columbia.edu)

37th Conference on Neural Information Processing Systems (NeurIPS 2023) Track on Datasets and Benchmarks.

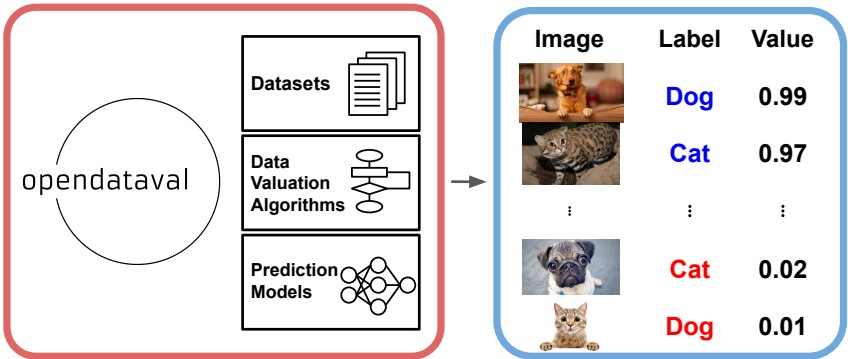

Figure 1: `OpenDataVal` is an open-source, easy-to-use and unified benchmark framework for data valuation. It contains a diverse collection of datasets, implementations of eleven state-of-the-art data valuation algorithms, and a variety of modern prediction models. In addition, `OpenDataVal` provides several downstream tasks for evaluating the quality of data valuation algorithms.

marketplace [1, 36]. BetaShapley generalizes DataShapley by relaxing the efficiency axiom in the Shapley value and has demonstrated its efficacy in identifying mislabeled images in the CIFAR-100 test dataset [17]. Recently, Ilyas et al. [11] propose datamodels and provide qualitative analyses of the train-test leakage that could hinder fair evaluations. Apart from these methods, many data valuation algorithms have been developed to capture different aspects of data values [42, 18] or improve computational efficiency [40]. Nevertheless, there has been little attention to establishing a user-friendly and standardized benchmarking system, and this is the main goal of this work.

**Our contributions** We introduce `OpenDataVal`, an easy-to-use and unified benchmark framework that allows researchers and practitioners to apply and compare different data valuation algorithms with a few lines of Python codes. `OpenDataVal` provides an integrated environment that includes (i) a diverse collection of image, natural language, and tabular datasets that are publicly available at open platforms OpenML [37], scikit-learn [27], and PyTorch [26], (ii) implementations of eleven different state-of-the-art data valuation algorithms in Table 1, and (iii) a prediction model API that enables to import any machine learning models in scikit-learn and PyTorch-based neural network models. In addition, `OpenDataVal` provides four downstream machine learning tasks for evaluating the quality of data values, noisy label data detection, noisy feature data detection, point removal experiment, and point addition experiment. We also demonstrate how `OpenDataVal` can be employed through benchmarking analysis in Section 4. We find no single data valuation algorithm performs uniformly superior across all the benchmark tasks. That is, different algorithms have their own advantages and capture different aspects of data values. We encourage users to choose an appropriate algorithm for their analysis.

`OpenDataVal` is designed to advance the transparency and reproducibility of data valuation works and to foster user engagement in data valuation. To support this endeavor, we have made `OpenDataVal` open-source at `https://github.com/opendataval/opendataval`, enabling anyone to contribute to our GitHub repository. Additionally, we have developed a leaderboard where users can participate in noisy data detection competitions. `OpenDataVal` will be regularly updated and maintained by the authors and participants.

**Related works** Sim et al. [33] conducted a technical survey of data valuation algorithms, focusing on the necessary assumptions and their desiderata. While their analysis provides mathematical insights, it lacks a comprehensive empirical comparison. Our framework primarily focuses on this problem, creating an easy-to-use benchmarking system that facilitates empirical comparisons. Valda[3] is a Python package with a set of data valuation algorithms including LOO, DataShapley [7], and BetaShapley [17]. `OpenDataVal` provides most of the data valuation algorithms available in Valda, plus additional methods, as well as an easy-to-use environment in which practitioners

---

[3]`https://uvanlp.org/valda/`

| Algorithms | Underlying Method | Quality of Data Values | | | |
|---|---|---|---|---|---|
| | | Noisy Label Detection | Noisy Feature Detection | Point Removal | Point Addition |
| LOO | Marginal contribution | - | - | + | + |
| DataShapley [7] | Marginal contribution | + | + | ++ | ++ |
| KNNShapley [12] | Marginal contribution | + | + | ++ | ++ |
| Volume-based Shapley [41] | Marginal contribution | - | - | - | - |
| BetaShapley [17] | Marginal contribution | + | + | ++ | ++ |
| DataBanzhaf [39] | Marginal contribution | - | - | + | + |
| AME [22] | Marginal contribution | - | - | ++ | + |
| InfluenceFunction [5] | Gradient | - | - | + | + |
| LAVA [13] | Gradient | - | ++ | + | + |
| DVRL [42] | Importance weight | + | - | + | ++ |
| Data-OOB [18] | Out-of-bag estimate | ++ | + | - | ++ |

Table 1: A taxonomy of data valuation algorithms available in `OpenDataVal`. LOO, KNN Shapley, Volume-based Shapley, DataShapley, BetaShapley, DataBanzhaf, and AME can be expressed as functions of marginal contributions. There have been alternative approaches to defining data values: the gradient, importance weights, and out-of-bag estimate. KNNShapley only works for KNN predictors; the other data valuation methods are compatible with any prediction models. The 'Quality of Data Values' section summarizes the benchmark results in Section 4. The symbols '- / + / ++' indicate that a corresponding data valuation method achieves a 'similar or worse / better / much better' performance than a random baseline, respectively. There is no single method that uniformly outperforms others in every task. Therefore, it is critical for users to selectively consider data valuation algorithms for their own downstream tasks.

and data analysts compute and evaluate various state-of-the-art data valuation algorithms in diverse downstream tasks.

DataPerf [24] has provided six realistic benchmark tasks: training set creation, test set creation, selection algorithm, debugging algorithm, slicing algorithm, and valuation algorithm. These tasks cover important data-centric problems that can arise in the model development process, but we emphasize that our main objective differs from that of DataPerf work. While they mainly focus on creating benchmark tasks and competitions, our main objective is to provide a unified environment for quantifying and comparing various data valuation algorithms.

## 2 A taxonomy of data valuation methods

In this section, we introduce a taxonomy of data valuation methods. For $d \in \mathbb{N}$, we denote an input space and an output space by $\mathcal{X} \subseteq \mathbb{R}^d$ and $\mathcal{Y} \subseteq \mathbb{R}$, respectively. We denote a training dataset by $\mathcal{D} = \{(x_i, y_i)\}_{i=1}^n$ where $x_i \in \mathcal{X}$ and $y_i \in \mathcal{Y}$ are an input and its label for the $i$-th datum. Then, all data valuation methods can be seen as a mapping that assigns a scalar score to each sample in $\mathcal{D}$, and each quantifies the impact of the point on the performance of a model trained on $\mathcal{D}$.

For instance, the leave-one-out (LOO) is defined as $\phi_{\text{LOO}}(x_i, y_i) := U(\mathcal{D}) - U(\mathcal{D} \backslash \{(x_i, y_i)\})$ where $U$ represents a preset utility function that takes as input a subset of the training dataset. In classification settings, a common choice for $U$ is the test accuracy of a model trained on the input. In this case, the $i$-th LOO value $\phi_{\text{LOO}}(x_i, y_i)$ measures the change in the test accuracy when the $i$-th data point $(x_i, y_i)$ is removed from the training procedure.

This concept of LOO has been generalized to the marginal contribution, which captures the average change in utility when a certain datum is removed from a set with a given cardinality. To be more specific, for a given cardinality $1 \le j \le n$, the marginal contribution of $(x_i, y_i) \in \mathcal{D}$ with respect to $j$ samples is defined as $\Delta_j(x_i, y_i) := \binom{n-1}{j-1}^{-1} \sum_{S \subseteq \mathcal{D}_j^{\backslash (x_i, y_i)}} U(S \cup \{(x_i, y_i)\}) - U(S)$ where $\mathcal{D}_j^{\backslash (x_i, y_i)} := \{S : S \subseteq \mathcal{D} \backslash \{(x_i, y_i)\}, |S| = j - 1\}$. Many data valuation methods are expressed as a function of marginal contributions, for instance, DataShapley [7] considers the simple average of marginal contributions $\phi_{\text{shap}}(x_i, y_i) := \frac{1}{n} \sum_{j=1}^n \Delta_j(x_i, y_i)$ and BetaShapley [17] considers a weighted average of marginal contributions $\phi_{\text{beta}}(x_i, y_i) := \sum_{j=1}^n w_{\text{beta},j} \Delta_j(x_i, y_i)$ for some

weight vector $(w_{\text{beta},1}, \ldots, w_{\text{beta},n})$. LOO, DataBanzhaf [39], DataModels [11], and the average marginal effect (AME) [22] are also included in this category, and they differ in how to combine marginal contributions into a single score.

There are alternative approaches to describing the concept of data values: gradient, importance weight, and out-of-bag estimate. The gradient methods are to quantify the rate at which a utility value changes when a particular data point is more weighted. The influence function [5] and LAVA [13] are included in this category. They differ depending on what utility value is being measured: the influence function uses the validation accuracy as a utility while LAVA uses the optimal transport cost.

The importance weight method learns a weight function $w : \mathcal{X} \times \mathcal{Y} \to \mathbb{R}$ so that an associated model $f_w$ achieves a good accuracy on the holdout validation dataset. Here, the model $f_w : \mathcal{X} \to \mathbb{R}$ is often obtained by minimizing a weighted risk $\mathbb{E}[w(X, Y)\ell(Y, f(X))]$ for some loss function $\ell : \mathcal{Y} \times \mathcal{Y} \to \mathbb{R}$ and a preset weight function $w$. Data valuation using reinforcement learning (DVRL) [42] is in this category, and they proposed a reinforcement learning technique to obtain a weight function.

An out-of-bag (OOB) estimation approach measures the contribution of each data point to out-of-bag accuracy when a bagging model is employed. Data-OOB [18] is in this category, and it captures how much a nominal label differs from predictions generated by weak learners. We provide a rigorous definition of each data valuation method along with their properties in Appendix. A taxonomy of data valuation algorithms with a brief summary of empirical results is presented in Table 1.

## 3 Overview of `OpenDataVal`

`OpenDataVal` provides a unified API called `ExperimentMediator`, which enables the quantification of data values and systemic comparative experiments. With `ExperimentMediator`, users can specify various datasets, data valuation algorithms, and hyperparameters (*e.g.*, a prediction model, model performance metric, and synthetic noise). The following code snippet illustrates how to compute data values using the embedding of the CIFAR10 dataset. Here, the embedding is obtained from the penultimate layer of the ResNet50 model [10] pretrained on the ImageNet dataset [4].

```python
from opendataval.experiment import ExperimentMediator
exper_med = ExperimentMediator.model_factory_setup(
    dataset_name="cifar10-embedding", model_name="ClassifierMLP",
    metric_name="accuracy", output_dir="outputs/",
    add_noise="mix_labels", noise_kwargs={"noise_rate": 0.2}
)
data_evaluators = [DataShapley(), DVRL()]
exper_med = exper_med.compute_data_values(data_evaluators)
```

Code snippet 1: This code snippet shows how to compute DataShapley [7] and DVRL [42] values for the CIFAR-10 dataset using a multilayer perceptron classifier as a prediction model, with classification accuracy as the evaluation metric. With 'add_noise' and 'noise_kwargs' arguments, a synthetic noise has been applied: 20% of the training dataset is randomly chosen and their original labels have been changed to different labels.

In addition, `OpenDataVal` provides a collection of evaluation tasks as demonstrated in Section 4. Using the computed data values in the previous step, users can conduct noisy label data detection tasks to systematically evaluate the performance of data valuation algorithms.

`OpenDataVal` provides a convenient way to compute and compare various state-of-the-art data valuation algorithms with a few lines of Python codes. It is also designed to be highly extensible and easily adaptable to custom experiment settings. We provide details on customization in Appendix.

**Leaderboards** `OpenDataVal` introduces the public leaderboards to promote transparency and healthy competition in the field of data valuation. In the initial release, we provide a noisy label data detection task for a selection of datasets. The challenge datasets, accessible within the `OpenDataVal` platform, contain label noise that is unknown to the user. The goal is to develop their own data valuation algorithms and find which data points have had noise added. Users can submit their

```
from opendataval.experiments import noisy_detection

# perform noisy label data detection and save results
exper_med.evaluate(noisy_detection, save_output=True)
>>>                  kmeans_f1
>>> DataShapley()      0.5625
>>> DVRL()             0.6825
```

Code snippet 2: This code snippet showcases the task of noise label data detection and evaluates the F1-scores of the DataShapley and DVRL algorithms.

| Dataset | Sample Size | Input Dimension | Number of Classes | Minor Class Proportion | Data Type | Source |
|---------|-------------|-----------------|-------------------|------------------------|-----------|--------|
| electricity | 38474 | 6 | 2 | 0.5 | Tabular | [6] |
| fried | 40768 | 10 | 2 | 0.498 | Tabular | OpenML-901 |
| 2dplanes | 40768 | 10 | 2 | 0.499 | Tabular | OpenML-727 |
| pol | 15000 | 48 | 2 | 0.336 | Tabular | OpenML-722 |
| MiniBooNE | 72998 | 50 | 2 | 0.5 | Tabular | [29] |
| nomao | 34465 | 89 | 2 | 0.285 | Tabular | [3] |
| bbc-embedding | 2225 | 768 | 5 | 0.17 | Text | [8] |
| IMDB-embedding | 50000 | 768 | 2 | 0.5 | Text | [23] |
| CIFAR10-embedding | 50000 | 2048 | 10 | 0.1 | Image | [16] |

Table 2: A summary of classification datasets used in the benchmarking analysis. For natural language and image datasets, the pretrained DistilBERT [30] and ResNet50 [10] models are employed to extract an embedding. For 'fried', '2dplanes', and 'pol' datasets, the number indicates their OpenML-Data-ID. All datasets are readily available in `OpenDataVal`.

algorithm outputs in the form of a CSV file, and our interactive ranking system will periodically update the leaderboard, showcasing the performance of different submissions. Submissions and competition details are available at `https://opendataval.github.io/leaderboards.html`.

## 4 Benchmarking Analysis

We present the practical applications of `OpenDataVal` through benchmarking analysis. We conduct the four machine learning tasks: noisy label data detection, noisy feature data detection, point addition experiment, and point removal experiment. These experiments have been widely used in the literature [7, 42, 13]. We provide details on evaluation metrics in Appendix. All the datasets, data valuation algorithms, prediction models, and downstream tasks employed in the benchmarking analysis are readily accessible within `OpenDataVal`, ensuring the transparency and reproducibility of all benchmark results.

**Experimental settings** We use the nine datasets in Table 2. These datasets are standard and widely used within the literature [7, 18]. Due to space limitations in the main text, we primarily focus on the analysis of the six tabular datasets, but our main findings are consistently observed across different modalities and datasets. Results for the text and image datasets are provided in Appendix. We compare the eleven different data valuation algorithms in Table 1. Additionally, we include a baseline algorithm, which assigns random numbers to data values. The InfluenceFunction, Volume-based Shapley, DataBanzhaf, AME, and Data-OOB require training models multiple times, and the number of trained models has a substantial impact on both performance and runtime. To ensure a fair comparison, we have set this parameter to a fixed number of $1000$. As for the other algorithms that do not require training models multiple times, we maintain fair comparisons by utilizing hyperparameters similar to those specified in the original papers. As for the base prediction model, we use a logistic regression model. However, it is important to note that the logistic regression model is not compatible with KNNShapley as it is implicitly based on KNN predictors and has a closed-form expression. The sample size for the training dataset is set to $n = 1000$. Implementation details are provided in Appendix.

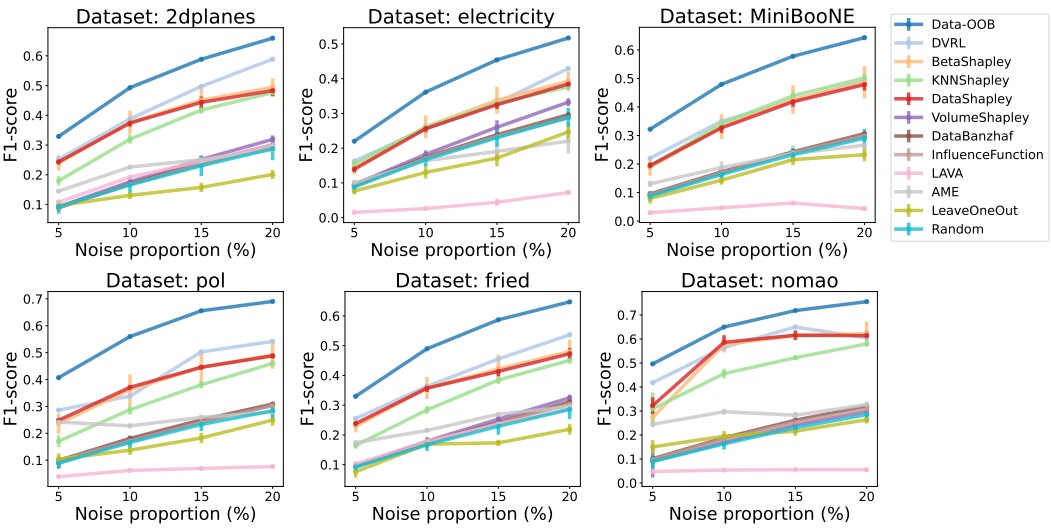

Figure 2: **Noisy label data detection.** F1-score of different data valuation algorithms on the four noise proportion settings. The higher the F1-score is, the better the data valuation algorithm is. The error bar indicates a 95% confidence interval based on 50 independent experiments. Data-OOB demonstrates significantly superior performance in detecting mislabeled data points in various situations.

## 4.1 Noisy data detection

We compare the detection capabilities of different data valuation algorithms using synthetically generated noisy datasets. We consider two types of synthetic noise. The first type is label noise where we flip the original label to its opposite label. The second type is feature noise where we add standard Gaussian random errors to the original features. After selecting one of two noises, we then randomly choose $p_{\mathrm{noise}}\%$ of the training dataset to perturb. Here, the four different levels of noise proportion $p_{\mathrm{noise}} \in \{5, 10, 15, 20\}$ is considered.

Data values are computed for this noisy dataset, and we investigate which data valuation algorithm is the most effective to identify these noisy data points. Based on the literature [39, 17, 18], we follow the approach of applying the $k$-means clustering algorithm to the data values [2]. This allows us to dichotomize the training dataset into two groups: beneficial and detrimental. We assign the detrimental label to the cluster with a lower average of data values. Given our expectation that mislabeled data points are likely to have low values, we predict data points within the detrimental group as noisy samples. We evaluate the F1-score between this prediction and the ground-truth annotation. Note that we do not use ground-truth annotations for noisy data points when computing data values. We only use them when we evaluate the quality of data valuation algorithms.

**Noisy label data detection** Figure 2 shows the F1-score of data valuation algorithms for the noisy label data detection tasks. The results can be grouped into three clusters based on their performance. The first cluster consists of a single algorithm, Data-OOB. It achieves significantly better results than other data valuation algorithms on most of the datasets and noise proportions. For instance, on the 'MiniBooNE' dataset with $p_{\mathrm{noise}} = 20\%$, Data-OOB shows $0.65$ F1-score when the second best algorithm KNNShapley shows $0.53$ and the random baseline shows $0.29$. The second cluster consists of DVRL, KNNShapley, DataShapley, and BetaShapley. They show reasonably sound detection ability compared to the random baseline. Volume-based Shapley, LAVA, LOO, InfluenceFunction, AME, and DataBanzhaf belong to the third cluster and show similar performance to the random baseline.

**Noisy feature data detection** Figure 3 shows the F1-score of different data valuation algorithms for the noisy feature data detection tasks. In contrast to the noisy label data detection, LAVA generally outperforms other methods with a substantial performance gap. This difference is potentially due to the transport cost function, one of the key hyperparameters in LAVA [13]. The default choice is designed to be sensitive to noises in features, leading to a good performance in noisy feature

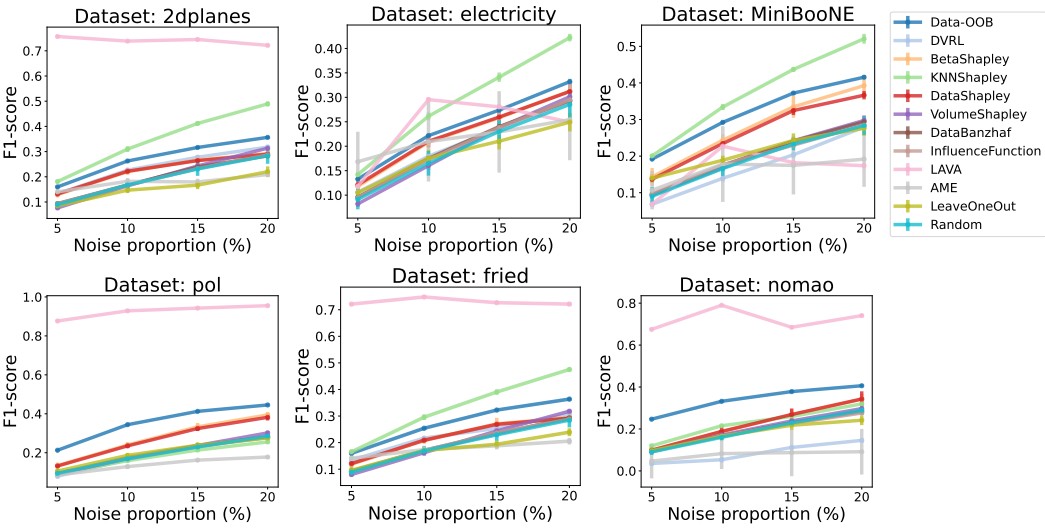

Figure 3: **Noisy feature data detection.** F1-score of different data valuation algorithms on the four noise proportion settings. The higher the F1-score is, the better the data valuation algorithm is. The error bar indicates a 95% confidence interval based on 50 independent experiments. LAVA generally outperforms other data valuation algorithms with a big performance gap.

data detection. However, we observe that LAVA's performance varies a lot across different datasets, suggesting that this choice of the transport cost function is important. Following LAVA, Data-OOB and Shapley-based methods demonstrate a better performance than the random baseline, showing their potential efficacy in noisy feature data detection. Volume-based Shapley, DVRL, InfluenceFunction, AME, and LOO show only comparable performance to the random baseline in this task.

## 4.2 Point removal and addition experiments

Data values can be used to find influential samples. To evaluate this quantitatively, we perform the point addition and removal experiments used in [7, 18]. The point removal experiment is performed with the following steps: For each data valuation algorithm, we remove data points from the entire training dataset in descending order of the data values. Each time the datum is removed, we fit a logistic regression model with the remaining dataset and evaluate its test accuracy on the holdout dataset. As we remove the data points in descending order, in the ideal case we remove the most helpful data points first, and thus model accuracy is expected to decrease. For the point addition experiment, we perform a similar procedure but add data points in ascending order. Similar to the point removal experiment, the model accuracy is expected to be low as we add detrimental data points first. Throughout the experiments, we used a perturbed dataset with label noise. The noise ratio is set to $p_{\mathrm{noise}} = 20\%$. The sample size of the holdout test dataset is set to 3000. All the procedures here can be easily performed in `OpenDataVal`.

**Point removal experiment**    Figure 4 shows test accuracy curves for the point removal experiment. Overall, all data valuation algorithms except Data-OOB perform better than the random baseline. Data-OOB often shows worse performance than the random baseline, indicating its effectiveness for finding low-quality data points does not imply finding helpful data points. AME, DataShapley, BetaShapley, and KNNShapley generally show superior performance. In particular, on the 'fried' dataset, these methods achieve at most 75% test accuracy after removing 20% of the most influential data points. DataBanzhaf and InfluenceFunction also show solid performance, while DVRL shows large performance differences across different datasets. LOO shows a slightly better performance than the random baseline.

**Point addition experiment**    Figure 5 shows test accuracy curves for the point addition experiment. All data valuation algorithms generally outperform the random baseline, showing their efficacy in identifying low-quality data points. Data-OOB uniformly shows the lowest test accuracy compared

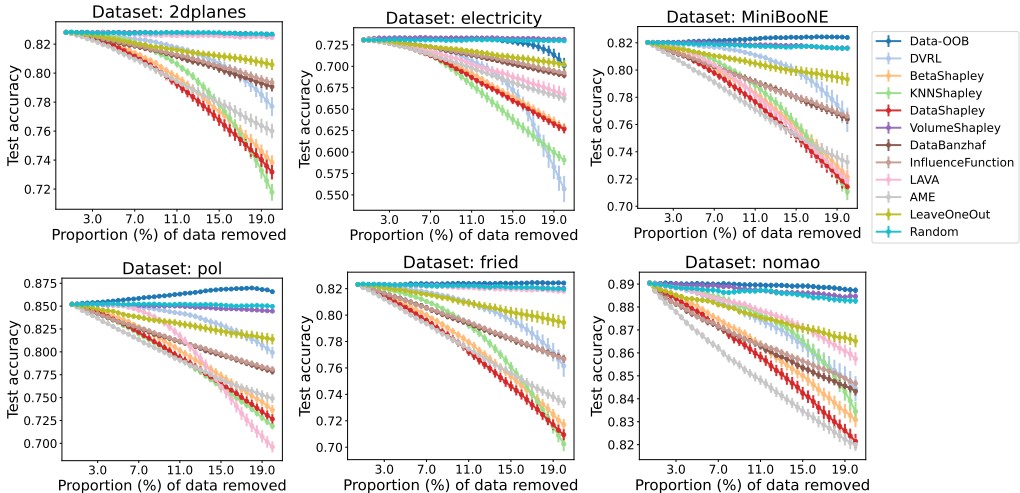

Figure 4: **Point removal experiment.** Test accuracy curves as a function of the most valuable data points removed. Lower curve indicates better data valuation algorithm. The error bar indicates a 95% confidence interval based on 50 independent experiments. AME and Shapley-based methods exhibit superior performance.

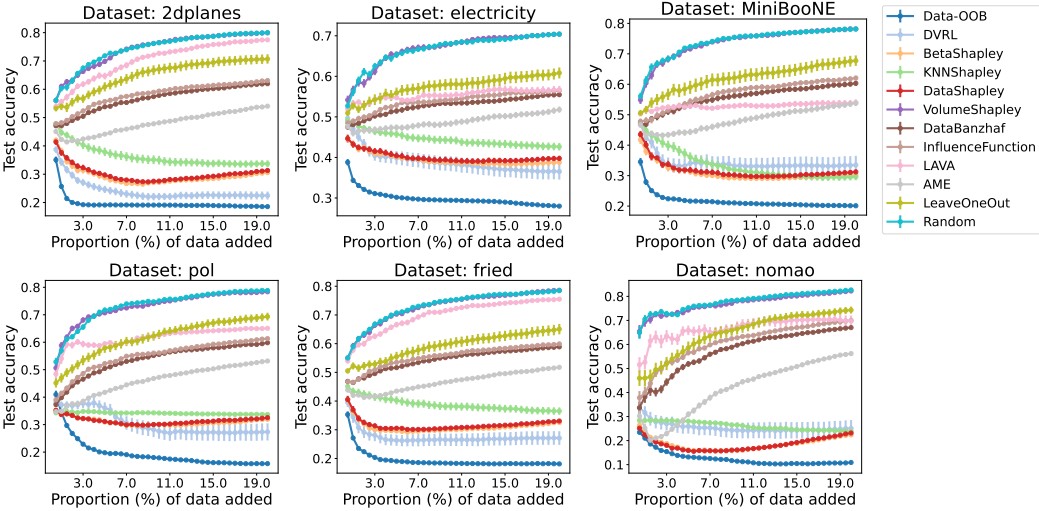

Figure 5: **Point addition experiment.** Test accuracy curves as a function of the least valuable data points added. Lower curve indicates better data valuation algorithm. The error bar indicates a 95% confidence interval based on 50 independent experiments. Data-OOB, DVRL, and Shapley-based methods exhibit superior performance.

to other algorithms, demonstrating the best performance in finding low-quality samples. Following Data-OOB, DVRL, DataShapley, and BetaShapley show promising performance in this task. Volume-based Shapley, LAVA, LOO, and InfluenceFunction show similar test accuracy, but InfluenceFunction exhibits a slightly better performance than other methods across all datasets. Volume-based Shapley is generally worse or only better than the random baseline, which has also been observed in the literature [13]. This is because the Volume-based Shapley is independent of the label information, and thus it incompletely reflects the quality of data points.

### 4.3 Runtime comparison

We also conduct a comparison of the runtime required to compute 1000 data values for the noisy label data detection task. In this task, we use the three tabular datasets (2dplanes, electricity, and

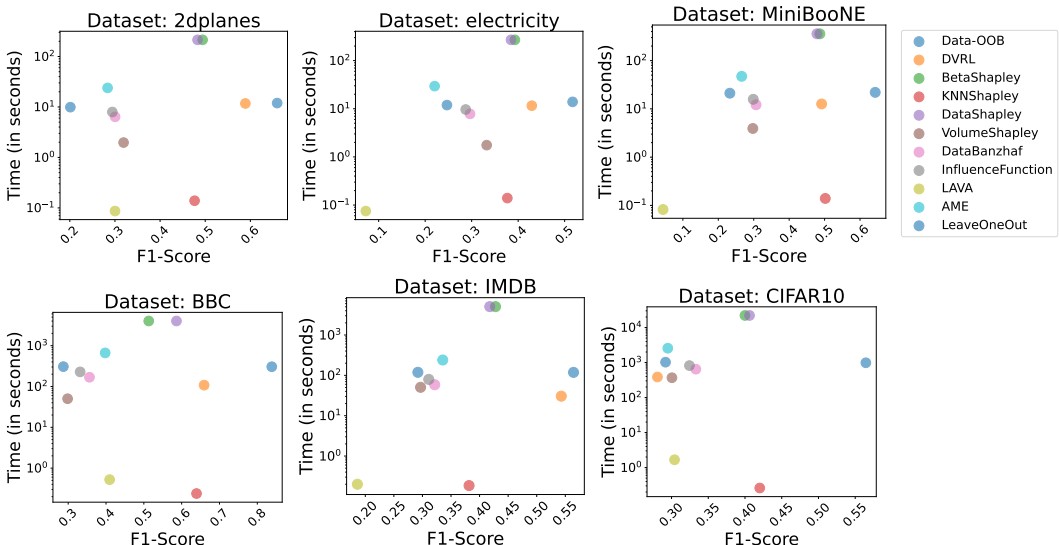

Figure 6: **Runtime comparison.** The runtime to compute 1000 data values is measured for the noisy label data detection task. The $x$-axis represents the F1-score used in Section 4.1 and the $y$-axis represents the runtime in the log scale. KNNShapley shows the best performance in terms of speed, while Data-OOB achieves a better F1-score in a reasonable time frame. Both DataShapley and BetaShapley are computationally expensive because they are based on marginal contribution estimation algorithms. They perform similarly in BBC, IMDB, and CIFAR10 datasets, and their points are visually overlapped.

MiniBooNE), two text datasets (BBC and IMDB), and one image dataset (CIFAR10) in order to show that findings are overall consistent across different modalities. We find a similar pattern with other tabular datasets. We utilize a noise dataset containing label noise with a noise ratio of $p_{\text{noise}} = 20\%$, but we highlight that the runtime is independent of the specific type of noise or its intensity. Figure 6 illustrates the runtime of the data valuation algorithms, along with the corresponding F1-scores for the noisy label data detection. Across most datasets, LAVA and KNNShapley demonstrate the shortest computation time because LAVA does not require model training and KNNShapley has a closed-form expression. In contrast, DataShapley and BetaShapley are shown to be computationally costly because they need to estimate the marginal contribution, which requires a large number of model training. Aside from these three algorithms, most of the other data valuation algorithms exhibit similar computational times as their key hyperparameter, the number of models to train, is set to the same value of 1000. Among them, Data-OOB demonstrates the best F1-score in noisy label data detection.

## 5   Discussion

In this paper, we propose `OpenDataVal`, an easy-to-use benchmark environment, and provide a convenient way to compute and compare data valuation algorithms with a few lines of Python codes. To ensure transparency and reproducibility, we have made our framework publicly available.

Our benchmarking analysis demonstrates no single method is uniformly superior across all evaluation metrics. Data-OOB achieves a powerful performance in the noisy label data detection tasks, but it performs less well in the point removal experiment. That is, it is very effective in identifying low-quality data points, but it fails to find the most positively influential data points. In contrast, AME shows the opposite results. By leveraging the LASSO model, it shows competitive performance in identifying the most beneficial data points in the point removal experiment, while it shows weak performance in finding low-quality data points in the noisy data detection task. We also find that Shapley-based methods (DataShapley, BetaShapley, and KNNShapley) have comparable performances. KNNShapley is faster but is limited to using KNN predictors, while DataShapley and BetaShapley can flexibly work with other predictors. Many theoretically rigorous and empirically

comprehensive analyses are critical, and we believe `OpenDataVal` opens the first step and facilitates fair and reproducible analyses in various experimental settings.

There are several intriguing future directions. In many real-world applications data can be duplicated and a fraction of them can be hard-encoded with a certain value (*e.g.*, -9999 for missing data) [14]. Furthermore, data can contain incomplete information in survival analysis, semi-supervised learning, and weakly-supervised learning settings, raising questions about how to evaluate the influence of such data points. Developing data valuation for sequentially observed data is also an important direction. For instance, in time-series forecasting or multi-armed bandit problems, previously observed data can substantially affect the subsequent data points, which makes it difficult to deploy techniques in typical i.i.d. learning settings. Developing a notion of data values in these scenarios will be an interesting and potentially influential problem.

In a broader context, the practical implementation of data valuation can give rise to several economic and societal questions. In data marketplaces, for instance, data providers might duplicate, transform, or even maliciously modify data to maximize their business profits. Most of the existing data valuation algorithms are not designed to handle these types of attacks, and as a result, erroneously evaluated data values can negatively affect the establishment of reliable and transparent data marketplaces. Developing incentive-compatible data valuation is an interesting new direction [32, 28, 34]. Another potential concern is data security in distributed and federated learning scenarios. Data owners may hesitate to share their data with the main server due to privacy issues, especially if data owners are competing with each other [34]. Evaluating data points based on aggregated summaries (e.g., gradients) will be an interesting topic as existing data valuation algorithms require direct data access.

## Acknowledgements

The authors would like to thank all anonymous reviewers for their comments. We would like to thank Eunsil Seok and Ian Covert for their constructive feedback. We also acknowledge computing resources from Columbia University's Shared Research Computing Facility project, which is supported by NIH Research Facility Improvement Grant 1G20RR030893-01, and associated funds from the New York State Empire State Development, Division of Science Technology and Innovation (NYSTAR) Contract C090171, both awarded April 15, 2010.

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

# Appendix

The appendix provides technical details on data valuation algorithms (Section A) and the four downstream tasks (Section B) available in `OpenDataVal`. Additional benchmarking analysis results are also presented (Section C) and we observe consistent findings on different datasets. Lastly, we provide detailed descriptions on `OpenDataVal` APIs in (Section D).

## A Data valuation algorithms

This section elaborates on different data valuation algorithms available in `OpenDataVal`. While some notations have already been defined in Section 2, we present a comprehensive set of notations here. For $d \in \mathbb{N}$, we denote an input space and an output space by $\mathcal{X} \subseteq \mathbb{R}^d$ and $\mathcal{Y} \subseteq \mathbb{R}$, respectively. We denote a training dataset by $\mathcal{D} = \{(x_i, y_i)\}_{i=1}^n$ where $x_i \in \mathcal{X}$ and $y_i \in \mathcal{Y}$ are an input and its label for the $i$-th datum. For a set $S$, we denote its power set by $2^S$ and its cardinality by $|S|$. We set $[j] := \{1, \ldots, j\}$ for $j \in \mathbb{N}$, and $\mathbb{1}(A)$ is an indicator whose value is one if a statement $A$ is true and zero otherwise.

### A.1 Marginal Contribution-based methods

The marginal contribution-based methods are formally introduced, which include LOO, DataShapley [7], KNNShapley [12], Volume-based Shapely [41], BetaShapley [17], DataBanzhaf [39], and AME [22]. We first define the marginal contribution below.

**Definition A.1** (Marginal contribution). For a set function $U : 2^{\mathcal{D}} \to \mathbb{R}$ and $j \in [n]$, the marginal contribution of $(x_i, y_i) \in \mathcal{D}$ with respect to $j$ samples is defined as follows.

$$\Delta_j(x_i, y_i; U) := \frac{1}{\binom{n-1}{j-1}} \sum_{S \subseteq \mathcal{D}_j^{\backslash (x_i, y_i)}} U(S \cup \{(x_i, y_i)\}) - U(S),$$

where $\mathcal{D}_j^{\backslash (x_i, y_i)} := \{S : S \subseteq \mathcal{D} \backslash \{(x_i, y_i)\}, |S| = j - 1\}$.

The marginal contribution $\Delta_j(x_i, y_i; U)$ measures the average change in $U$ when a particular data point $(x_i, y_i)$ is removed. Here, a common choice for a set function $U$ is the performance of a model trained on a subset of the training dataset $\mathcal{D}$. We provide two examples below.

**Example A.1** (Classification). *A default choice for a set function $U$ in* `OpenDataVal` *is the validation classification accuracy of an empirical risk minimizer trained on a subset of $\mathcal{D}$. To be more specific, we denote the validation dataset by $\{(\tilde{x}_i, \tilde{y}_i)\}_{i=1}^{n_{\mathrm{val}}}$. Then,*

$$U(S) := \frac{1}{n_{\mathrm{val}}} \sum_{i=1}^{n_{\mathrm{val}}} \mathbb{1}(\tilde{y}_i = \hat{f}_S(\tilde{x}_i)),$$

*where $\hat{f}_S := \operatorname{argmin}_{f \in \mathcal{F}} \sum_{j \in S} \mathbb{1}(y_j \neq f(x_j))$ for some class of classification models $\mathcal{F} = \{f | f : \mathcal{X} \to \mathcal{Y}\}$. When $S = \{\}$ is the empty set, $U(S)$ is set to be the performance of the best constant model by convention.*

**Example A.2** (Regression). *A default choice for a set function $U$ in* `OpenDataVal` *is the validation negative mean squared error of an empirical risk minimizer trained on a subset of $\mathcal{D}$. Using the same notation defined in Example A.1,*

$$U(S) := -\frac{1}{n_{\mathrm{val}}} \sum_{i=1}^{n_{\mathrm{val}}} (\tilde{y}_i - \hat{f}_S(\tilde{x}_i))^2,$$

*where $\hat{f}_S := \operatorname{argmin}_{f \in \mathcal{F}} \sum_{j \in S} (y_j - f(x_j))^2$ for some class of regression models $\mathcal{F} = \{f | f : \mathcal{X} \to \mathcal{Y}\}$.*

It is noteworthy that a set function $U$ may depend on the choice of learning algorithms (*e.g.*, empirical risk minimization) and a model class $\mathcal{F}$. For notational simplicity, we suppress a learning algorithm and a model class.

**Estimation of the marginal contributions.** Following the existing literature [7], `OpenDataVal` uses a truncated Monte Carlo algorithm. Specifically, let $\pi = (\pi_1, \ldots, \pi_n) \in \mathbb{N}^n$ be a random permutation of $[n]$. Then, for $j \in [n]$, we compute the utility $U(\{(x_{\pi_k}, y_{\pi_k})\}_{k=1}^j)$. To gradually increase the cardinality, we iteratively increment $j$ from 1 to $n$. We stop increasing the cardinality if a relative change is converged. Specifically, for $l \in [n]$, we define the relative change as follows.

$$V_l := \frac{|U\left(\{(x_{\pi_k}, y_{\pi_k})\}_{k=1}^l \cup \{x_{\pi_{l+1}}, y_{\pi_{l+1}}\}\right) - U\left(\{(x_{\pi_k}, y_{\pi_k})\}_{k=1}^l\right)|}{U\left(\{(x_{\pi_k}, y_{\pi_k})\}_{k=1}^l\right)}.$$

We stop increasing the cardinality at $j = \mathrm{argmin}\{j' \in [n] : |\{l \leq j' : V_l \leq 10^{-8}\}| \geq 10\}$, *i.e.*, it is the smallest cardinality $j$ that there are 10 cardinalities less than $j$ with a small relative change $V_l$.

While repeatedly generating random permutations, we use the utility gap to estimate the marginal contribution. In other words, the multiple estimates of $U(\{\pi_1, \ldots, \pi_j, \pi_{j+1}\}) - U(\{\pi_1, \ldots, \pi_j\})$ are used to estimate $\Delta_{j+1}(x_{\pi_{j+1}}, y_{\pi_{j+1}})$. We stop generating random permutations when a simple average of marginal contributions is converged. We used the Gelman-Rubin statistics [38] by constructing ten independent Monte Carlo chains for each training data point. The maximum of these statistics across samples is considered, and we stop drawing random permutations if the maximum value of Gelman-Rubin statistics is less than the threshold value of $1.05$ or the number of random permutations exceeds $10^4$. In `OpenDataVal`, to obtain reliable estimates, we set the default value for the minimum number of random permutations as $300$.

**LOO.** The LOO measures the change in $U$ when one data point of interest is removed from the entire dataset, and it is given as follows.

$$\phi_{\mathrm{loo}}(x_i, y_i) := \Delta_n(x_i, y_i).$$

Note that $\phi_{\mathrm{loo}}(x_i, y_i) = U(\mathcal{D}) - U(\mathcal{D}\backslash\{(x_i, y_i)\})$. In `OpenDataVal`, the computation of LOO is done by naively computing $U(\mathcal{D})$ and $U(\mathcal{D}\backslash\{(x_i, y_i)\})$ for all $i \in [n]$.

**DataShapley.** DataShapley takes a simple average of all the marginal contributions as follows.

$$\phi_{\mathrm{shap}}(x_i, y_i) := \frac{1}{n}\sum_{j=1}^n \Delta_j(x_i, y_i).$$

In `OpenDataVal`, as suggested in Ghorbani and Zou [7], we estimate the marginal contributions using the truncated Monte Carlo algorithm and compute its simple average.

**KNNShapley.** KNNShapley, like DataShapley, is based on the Shapley value but is characterized by the use of a specific utility function. In classification settings, for instance, KNNShapley uses the following utility function.

$$U(S) = \frac{1}{n_{\mathrm{val}}k}\sum_{i=1}^{n_{\mathrm{val}}}\sum_{(x_j, y_j) \in \mathcal{N}(\tilde{x}_i, S)} \mathbb{1}(\tilde{y}_i = y_j),$$

where $k \in [n]$ is a preset hyperparameter and represents $k$ nearest neighbors in KNN predictors and $\mathcal{N}(\tilde{x}_i, S) \subseteq \mathcal{D}$ is a set of $\min(k, |S|)$ nearest neighbors of $\tilde{x}_i$. In regression settings,

$$U(S) = -\frac{1}{n_{\mathrm{val}}k}\sum_{i=1}^{n_{\mathrm{val}}}\sum_{(x_j, y_j) \in \mathcal{N}(\tilde{x}_i, S)} (\tilde{y}_i - y_j)^2.$$

With these utility functions, [12] showed that KNNShapley yields a closed-form expression, and they are implemented in `OpenDataVal`.

**Volume-based Shapley** The idea of the volume-based Shapley by Xu et al. [41] is to use the same Shapley value function as DataShapley, but it is characterized by using the volume of input data for a utility function.

$$U(S) = \mathrm{Vol}(X_S) := \sqrt{|\sum_{i \in S} x_i x_i^T|}, \tag{1}$$

where $|G|$ is a determinant of a matrix $G \in \mathbb{R}^{d \times d}$. It is noteworthy that Equation (1) does not require a validation dataset, leading to a validation-free data valuation method. However, it does not use label information, and thus it is not appropriate to describe the quality of labels. The previous work [13] observed that it is often the main reason for poor performance in many downstream tasks.

**BetaShapley.** BetaShapley has a form of a weighted mean of the marginal contributions.

$$\phi_{\text{beta}}(x_i, y_i) := \sum_{j=1}^{n} w_{\text{beta},j} \Delta_j(x_i, y_i),$$

where $w_{\text{beta},j} = \binom{n-1}{j-1} \frac{\text{Beta}(j+\beta-1, n-j+\alpha)}{\text{Beta}(\alpha, \beta)}$, $\text{Beta}(\alpha, \beta) = \Gamma(\alpha)\Gamma(\beta)/\Gamma(\alpha + \beta)$ is the Beta function and $\Gamma$ is the Gamma function. In `OpenDataVal`, $(\alpha, \beta) = (4, 1)$ is set to the default parameter[4], and similar to DataShapley, BetaShapley values are obtained after estimating the marginal contributions.

**DataBanzhaf.** DataBanzhaf is based on the Banzhaf value, which is defined as follows.

$$\phi_{\text{banz}}(x_i, y_i) := \sum_{j=1}^{n} w_{\text{banz},j} \Delta_j(x_i, y_i),$$

where $w_{\text{banz},j} = 2^{-n} \binom{n-1}{j-1}$. In `OpenDataVal`, instead of using the marginal contribution estimates, we follow the original algorithms proposed by Wang and Jia [39].

**AME.** Lin et al. [22] proposed the average marginal effect, which is also known as AME.

$$\psi_{\text{AME}}(x_i, y_i) := \mathbb{E}_S[U(S \cup \{(x_i, y_i)\}) - U(S)],$$

where the expectation $\mathbb{E}_S$ is taken over a random set $S$ with a user-defined distribution defined on the discrete space $\cup_{j=1}^{n} \mathcal{D}_j^{\setminus(x_i, y_i)}$. Lin et al. [22] showed that AME can be efficiently estimated by predicting a model's prediction. Specifically, for a regularization parameter $\lambda > 0$, a predefined transformation function $g : \{0, 1\}^n \to \mathbb{R}^n$, and a set $\mathcal{S} = \{S : S \subseteq \mathcal{D}\}$, AME can be estimated as a solution of a LASSO regression model.

$$\text{argmin}_{\gamma \in \mathbb{R}^n} \frac{1}{|\mathcal{S}|} \sum_{S \in \mathcal{S}} \left( U(S) - g(\mathbb{1}_S)^T \gamma \right)^2 + \lambda \sum_{i=1}^{n} |\gamma_i|,$$

where $\mathbb{1}_S \in \{0, 1\}^n$ is $n$-dimensional vector whose element is one if its index is an element of $S$, zero otherwise. In `OpenDataVal`, following Lin et al. [22], the same transformation function $g$ and the $\mathcal{S}$ that follows the same uniform distribution are used. We choose the optimal regularization parameter with the cross-validation using 'LassoCV' in 'scikit-learn' with its default parameter values.

### A.2 Gradient-based methods

**Influence function.** The influence function is originally introduced in robust statistics to measure the effect of a data point on a statistical estimator by its Gateaux derivative [9]. In machine learning, it has been often approximated by LOO, or more generally, the difference between two average model performances: one containing a data point of interest in the training procedure and the other not [15][5]. To be more specific, for a pre-defined hyperparameter $M \in [n]$, we denote a set of subsets with the same cardinality by $\mathcal{S}^{(M)} := \{S : S \subseteq \mathcal{D}, |S| = M\}$. For $i \in [n]$, we set $\mathcal{S}_{i,\text{in}}^{(M)} := \{S \in \mathcal{S}^{(M)} : (x_i, y_i) \in S\}$ and $\mathcal{S}_{i,\text{out}}^{(M)} := \{S \in \mathcal{S}^{(M)} : (x_i, y_i) \notin S\}$. Note that $\mathcal{S}_{i,\text{in}}^{(M)} \cup \mathcal{S}_{i,\text{out}}^{(M)} = \mathcal{S}$ and $\mathcal{S}_{i,\text{in}}^{(M)} \cap \mathcal{S}_{i,\text{out}}^{(M)} = \{\}$ for all $i \in [n]$. Then, the influence function used in Feldman and Zhang [5] is defined as follows.

$$\phi_{\text{IF}}(x_i, y_i) := \frac{1}{|\mathcal{S}_{i,\text{in}}^{(M)}|} \sum_{S \in \mathcal{S}_{i,\text{in}}^{(M)}} U(S) - \frac{1}{|\mathcal{S}_{i,\text{out}}^{(M)}|} \sum_{S \in \mathcal{S}_{i,\text{out}}^{(M)}} U(S).$$

In `OpenDataVal`, as suggested by Feldman and Zhang [5], $M$ is chosen as $\lfloor 0.7 \times n \rfloor$ where $\lfloor x \rfloor$ is the biggest integer that is less than or equal to $x$.

---

[4]The authors in [17] show that $(\alpha, \beta) = (16, 1)$ yields the best performance, but we empirically find $(\alpha, \beta) = (4, 1)$ also yields competitive results in various datasets. In our settings, $(\alpha, \beta) = (4, 1)$ often shows better results, and thus it is chosen as the default value.

[5]The authors in [15] proposed an efficient way to compute the influence function using Hessian-vector products, but it has been approximated with LOO.

**LAVA.** LAVA [13] proposed to measure how fast the optimal transport cost between a training dataset and a validation dataset changes when a training data point of interest is more weighted. This idea is formalized via the gradient of the optimal transport cost with respect to the probability mass, and in their paper, they showed that this gradient is expressed as follows.

$$\phi_{\text{LAVA}}(x_i, y_i) := h_i^* - \frac{1}{n-1} \sum_{j \in [n] \setminus \{i\}} h_j^*,$$

where $(h_1^*, \ldots, h_n^*)$ is a part of the optimal solution of the dual problem: for a validation dataset $\{(\tilde{x}_i, \tilde{y}_i)\}_{i=1}^{n_{\text{val}}}$,

$$h^*, g^* := \operatorname{argmax}_{h,g \in C^0(\mathcal{X} \times \mathcal{Y})} \langle h, \frac{1}{n} \delta_{(x_i, y_i)} \rangle + \langle g, \frac{1}{n_{\text{val}}} \delta_{(\tilde{x}_i, \tilde{y}_i)} \rangle,$$

where $C^0(\mathcal{X} \times \mathcal{Y})$ is the set of all continuous function defined on $\mathcal{X} \times \mathcal{Y}$, and $\delta_{(x,y)}$ is the Delta measure at $(x, y) \in \mathcal{X} \times \mathcal{Y}$.

### A.3 Importance weight-based methods

Data valuation using reinforcement learning (DVRL) proposed by Yoon et al. [42] involves the utilization of reinforcement learning algorithms to compute data values. DVRL trains a model $h : \mathcal{X} \times \mathcal{Y} \to [0, 1]$ that outputs its importance weight by solving the following objective function.

$$\min_{h \in \mathcal{H}} \mathbb{E}[(Y - f_h(X))^2] \quad \text{s.t.} \quad f_h = \operatorname{argmin}_{f \in \mathcal{F}} \mathbb{E}[h(X, Y)(Y - f(X))^2],$$

where $\mathcal{H} := \{h : \mathcal{X} \times \mathcal{Y} \to [0, 1]\}$ and $\mathcal{F} := \{f : \mathcal{X} \to \mathcal{Y}\}$. Then, the data value of $(x, y)$ is computed as the importance weight $h(x, y)$. In `OpenDataVal`, we follow the hyperparameters used in Yoon et al. [42].

### A.4 Out-of-bag estimate-based methods

Data-OOB by [18] is a distinctive data valuation algorithm, which uses the out-of-bag estimate to describe the quality of data. Suppose we have a bagging model that consists of $B$ weak learner models: for $b \in [B]$, we denote the $b$-th weak learner by $\hat{f}_b : \mathcal{X} \to \mathcal{Y}$, which is trained on the $b$-th bootstrap dataset. It can be expressed as a minimizer of a weighted risk as follows.

$$\hat{f}_b := \operatorname{argmin}_{f \in \mathcal{F}} \frac{1}{n} \sum_{j=1}^{n} w_{bj} \ell(y_j, f(x_j)),$$

where $\ell : \mathcal{Y} \times \mathcal{Y} \to \mathbb{R}$ is a loss function and $w_{bj} \in \mathbb{Z}$ is the number of times the $j$-th datum $(x_j, y_j)$ is selected in the $b$-th bootstrap dataset. With these notations, for $i \in [n]$, we propose to use the following quantity as data values.

$$\psi_{\text{OOB}}(x_i, y_i) := \frac{\sum_{b=1}^{B} \mathbb{1}(w_{bi} = 0) T(y_i, \hat{f}_b(x_i))}{\sum_{b=1}^{B} \mathbb{1}(w_{bi} = 0)}. \tag{2}$$

where $T : \mathcal{Y} \times \mathcal{Y} \to \mathbb{R}$ is a score function that represents the goodness of a weak learner $\hat{f}_b$ at the $i$-th datum $(x_i, y_i)$. For instance, we can use the correctness function $T(y_i, \hat{f}_b(x_i)) = \mathbb{1}(y_i = \hat{f}_b(x_i))$ in classification settings and the negative Euclidean distance $T(y_i, \hat{f}_b(x_i)) = -(y_i - \hat{f}_b(x_i))^2$ in regression settings.

## B  Tasks and their evaluation metrics

We formally introduce the four machine learning tasks that assess the effectiveness of data evaluation algorithms: (i) noisy label data detection, (ii) noisy feature data detection, (iii) point removal experiment, and (iv) point addition experiment.

## B.1 Noisy label data detection

The main goal of noisy label data detection is to identify data points that have been mislabeled. We consider a synthetically generated mislabeled dataset: we randomly choose $p_{\mathrm{noise}}\%$ of the training dataset and replace the original label with a different label. Let $S^{(\mathrm{mislabeled})} \subseteq [n]$ be a set of indices of mislabeled data points. This annotation is not available to any data valuation algorithms and is only available when we assess the data valuation algorithms.

Let $\phi : \mathcal{X} \times \mathcal{Y} \to \mathbb{R}$ be a data valuation algorithm and denote a set of computed data values by $\Phi_n := (\phi(x_1, y_1), \dots, \phi(x_n, y_n))$. We divide a set of $n$ points $\Phi_n$ into two clusters using the $k$-means clustering algorithm [2] and denote these two clusters of indices by $S_\phi^{(\mathrm{low})}$ and $S_\phi^{(\mathrm{high})}$. That is, $S_\phi^{(\mathrm{low})} \cup S_\phi^{(\mathrm{high})} = [n]$ and $S_\phi^{(\mathrm{low})} \cap S_\phi^{(\mathrm{high})} = \{\}$. Without loss of generality, we assume $S^{(\mathrm{low})}$ has a lower mean than $S_\phi^{(\mathrm{high})}$, *i.e.* $|S_\phi^{(\mathrm{low})}|^{-1} \sum_{i \in S_\phi^{(\mathrm{low})}} \phi(x_i, y_i) \leq |S_\phi^{(\mathrm{high})}|^{-1} \sum_{i \in S_\phi^{(\mathrm{high})}} \phi(x_i, y_i)$.

Since it is desirable for mislabeled data points to have lower values, we regard $S_\phi^{(\mathrm{low})}$ as the prediction for the set of mislabeled data points. Then, we compute the F1-score between the prediction set $S_\phi^{(\mathrm{low})}$ and the ground-truth annotation $\mathcal{S}^{(\mathrm{mislabeled})}$ as follows

$$\mathrm{F1}(S_\phi^{(\mathrm{low})}, \mathcal{S}^{(\mathrm{mislabeled})}) := 2 \frac{|S_\phi^{(\mathrm{low})} \cap \mathcal{S}^{(\mathrm{mislabeled})}|}{|S_\phi^{(\mathrm{low})}| + |\mathcal{S}^{(\mathrm{mislabeled})}|}.$$

It is noteworthy that $\mathrm{F1}(S_\phi^{(\mathrm{low})}, \mathcal{S}^{(\mathrm{mislabeled})})$ is the harmonic mean of the precision and recall as the precision is expressed as $\frac{|S_\phi^{(\mathrm{low})} \cap \mathcal{S}^{(\mathrm{mislabeled})}|}{|S_\phi^{(\mathrm{low})}|}$ and the recall is expressed as $\frac{|S_\phi^{(\mathrm{low})} \cap \mathcal{S}^{(\mathrm{mislabeled})}|}{|\mathcal{S}^{(\mathrm{mislabeled})}|}$.

## B.2 Noisy feature data detection

The main goal of noisy feature data detection is to identify data points with perturbed features. Similar to Section B.1, we consider a synthetically generated perturbed dataset: we randomly choose $p_{\mathrm{noise}}\%$ of the training dataset and add some Gaussian random errors to every original feature. Let $\mathcal{S}^{(\mathrm{perturbed})} \subseteq [n]$ be a set of indices of perturbed data points. For each data valuation algorithm $\phi$, we compute $S_\phi^{(\mathrm{low})}$, and then we evaluate $\mathrm{F1}(S_\phi^{(\mathrm{low})}, \mathcal{S}^{(\mathrm{perturbed})})$ as in Section B.1.

## B.3 Point removal experiment

Let $\phi : \mathcal{X} \times \mathcal{Y} \to \mathbb{R}$ be a data valuation algorithm and denote a set of computed data values by $\Phi_n := (\phi(x_1, y_1), \dots, \phi(x_n, y_n))$. Suppose $\pi = (\pi_1, \dots, \pi_n) \in \mathbb{N}^n$ represents indices of data values in descending order, *i.e.*, $\phi(x_{\pi_1}, y_{\pi_1}) \geq \cdots \geq \phi(x_{\pi_n}, y_{\pi_n})$. In the point removal experiment, we remove data points one by one in descending order from the entire training dataset. That is, we remove $(x_{\pi_1}, y_{\pi_1})$ first and $(x_{\pi_n}, y_{\pi_n})$ last. Each time the datum is removed, we fit a logistic model with the remaining dataset. For instance, after $k$ data points are removed, we train a logistic model with $\mathcal{D} \setminus \{(x_{\pi_j}, y_{\pi_j})\}_{j=1}^k$. We denote the performance after $k$ data points are removed by $\mathrm{perf}_k$. Then, for preset $K \in \mathbb{N}$, the point removal experiment evaluates the average performance after the removal of $K$ points:

$$\frac{1}{K} \sum_{k=1}^K \mathrm{perf}_k.$$

Throughout this paper, we consider $K = \lfloor 0.2 \times n \rfloor$, and the classification accuracy is used to compute the performance $\mathrm{perf}_k$. As we removed the most valuable data points first, a lower value indicates a better performance. The point removal curves in Figures 4 and 9 are $(\mathrm{perf}_1, \dots, \mathrm{perf}_K)$, and thus the evaluation metric represents the normalized area under the curve. In experiments, a coarser grid is used that evaluates $\mathrm{perf}$ every five-point removal, not one data point.

## B.4 Point addition experiment

The point addition experiment is similar to the point removal experiment, but the only difference is that we add data points from the empty set in ascending order. That is, we first add first $(x_{\pi_n}, y_{\pi_n})$

and $(x_{\pi_1}, y_{\pi_1})$ last. The evaluation metric for the point addition experiment is

$$\frac{1}{K} \sum_{k=1}^{K} \mathrm{perf}_{n-k}.$$

We consider $K = \lfloor 0.2 \times n \rfloor$, and the classification accuracy is used for the performance $\mathrm{perf}_k$. A lower value indicates a better performance. The curves in Figures 5 and 10 are $(\mathrm{perf}_{n-1}, \ldots, \mathrm{perf}_{n-K})$. See Section B.3 for more details.

## C  Additional Benchmarking Analysis

### C.1  Missing details on experimental settings

**Datasets**  For tabular datasets, a standard normalization procedure is employed, wherein each feature is centered and normalized to have a mean of zero and a standard deviation of one. However, for non-tabular datasets such as CIFAR10, BBC, and IMDB, no specific normalization procedures are applied except for the feature extraction. We use the pretrained ResNet50 model for CIFAR10 [10] and the pretrained DistilBERT model for texts [30]. Once the preprocessing steps are completed, following the literature [18], we split it into three datasets, namely, a training dataset, a validation dataset, and a test dataset. The data valuation process focuses on assessing the value of data within the training dataset, while the validation dataset is utilized for evaluating the utility function.[6]. The sample sizes for the training and validation datasets are set to 1000 and 100, respectively. The size of the test dataset is fixed at 3000 for all datasets, except for the text datasets (BBC and IMDB), where it is set to 500. The test dataset is used for point removal and addition experiments only when evaluating the test accuracy. As for the noisy feature data detection problem, we generate a random error from a Gaussian distribution with a mean of zero and a standard deviation of two. Although our benchmark analysis is fair and captures the gist of different data valuation algorithms, we encourage users to establish various experimental settings with OpenDataVal.

**Data valuation algorithms**  We compare the eleven different data valuation algorithms listed in Table 1. Regarding the hyperparameters, we use the default settings provided by OpenDataVal, with most of the default values derived from the original research papers. There are some hyperparameters we choose for a fair and efficient comparison: for the KNNShapley, we set the number of $k$-nearest neighbors to 10% of training data points. In the case of DVRL, the model is trained for a total of 2000 epochs. Data-OOB, InfluenceFunction, AME, and DataBanzhaf have a shared key parameter 'the number of models to train', and it is set to the same number of 1000, throughout all experiments.

### C.2  Additional Experimental Results

We show additional benchmarking analysis results using one image dataset (CIFAR10) and two text datasets (BBC and IMDB). Figures 7 and 8 show the noisy label and feature data detection results on the three non-tabular datasets, respectively. Figures 9 and 10 show the point removal and addition experiment results, respectively.

Overall, the findings presented in Section 4 are consistently observed in the non-tabular datasets. That is, Data-OOB shows promising empirical performance in detecting low-quality data points but turns out to be less effective in identifying beneficial data points. Shapley-based methods exhibit similar performance to each other, which is better than a random baseline. AME performs poorly in identifying low-quality data points, which is likely due to its sparse data values, but shows competitive performance in the point removal and addition experiments. InfluenceFunction and LOO outperform a random baseline in point removal/addition experiments, but they often perform worse than other data valuation algorithms.

### C.3  Robustness of findings against different prediction models

We conduct additional numerical experiments to demonstrate our empirical findings in Section 4 are robust against different prediction models. We compare the result with a logistic regression model

---

[6]Note that Data-OOB does not require the validation dataset and it does not use additional validation data points.

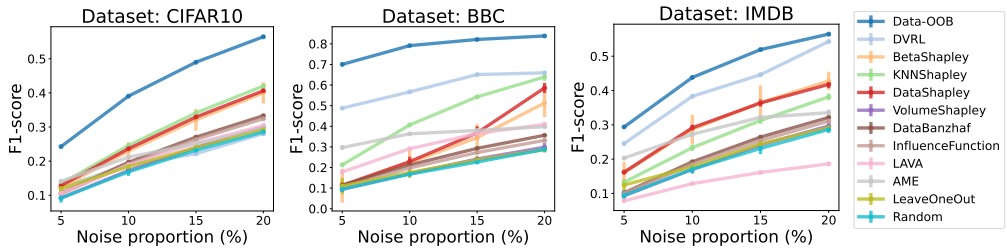

Figure 7: **Noisy label data detection on non-tabular datasets.** F1-score of different data valuation algorithms on the four noise proportion settings. The higher the F1-score is, the better the data valuation algorithm is. The error bar indicates a 95% confidence interval based on 50 independent experiments. Data-OOB significantly outperforms other methods in detecting mislabeled data points in various situations.

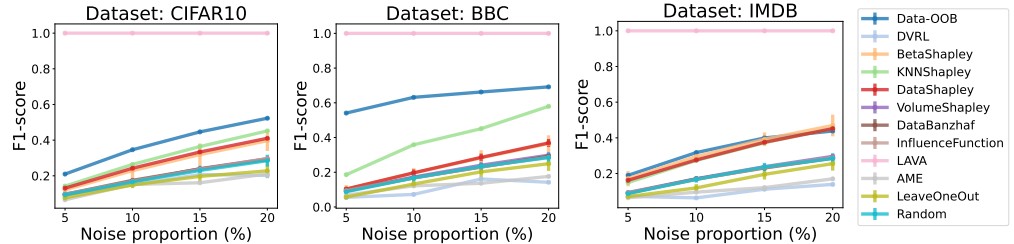

Figure 8: **Noisy feature data detection on non-tabular datasets.** F1-score of different data valuation algorithms on the four noise proportion settings. The higher the F1-score is, the better the data valuation algorithm is. The error bar indicates a 95% confidence interval based on 50 independent experiments. As in tabular datasets, LAVA shows remarkable performance compared to other data valuation algorithms.

with that of two prediction models: a decision tree model, denoted by TREE, and a support vector machine, denoted by SVM. As for the datasets, we used 2dplanes, pol, fried, and IMDB. We consider every data valuation algorithm used in Section 4, but we excluded DataShapley and BetaShapley for IMDB due to their expensive computational costs.

Figures 11 and 12 show that the F1-score comparison on the four datasets in noisy label data detection and noisy feature data detection, respectively. Note that the performances of both KNNShapley and LAVA do not vary across different prediction models as their algorithms do not rely on a prediction model. As for other data valuation algorithms, while there are some variations, we observe an overall consistent trend across different prediction models: Data-OOB (*resp.* LAVA) outperforms other methods with a large margin in the noisy label (*resp.* feature) data detection task. Another finding is that the performance of DVRL is highly variable compared to other methods. This could be because of its dependence on complicated models and the instability of the reinforcement learning algorithm.

Every data valuation algorithm has various hyperparameters that could affect the performance by a large margin. In our experiments, we followed the literature, and the default or suggested hyperparameters were employed. However, we acknowledge that it is possible that carefully tuned hyperparameters can yield better or different results. Comprehensive and extensive comparisons can be done with OpenDataVal, and it will be an important future research topic.

# D    Available APIs in `OpenDataVal`

## D.1    Datasets

We separate the preprocessing of a raw dataset (acquisition and parsing) and the preparation for a numerical experiment (splitting, noisifying) using a Register and DataFetcher object, respectively. A Register object downloads, caches, and preprocesses the raw dataset. The DataFetcher, which

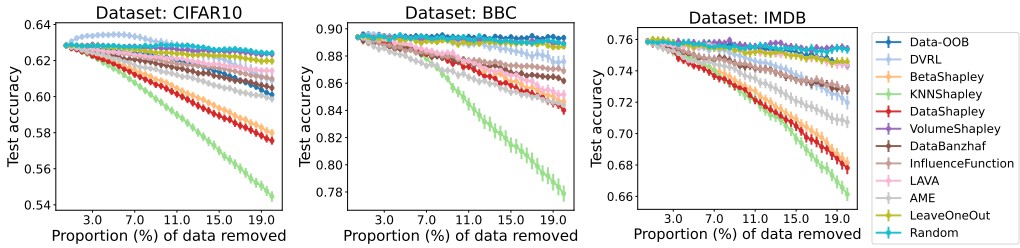

Figure 9: **Point removal experiment on non-tabular datasets.** Test accuracy curves as a function of the most valuable data points removed. Lower curve indicates better data valuation algorithm. The error bar indicates a 95% confidence interval based on 50 independent experiments. AME and Shapley-based methods, especially KNNShapley, exhibit superior performance.

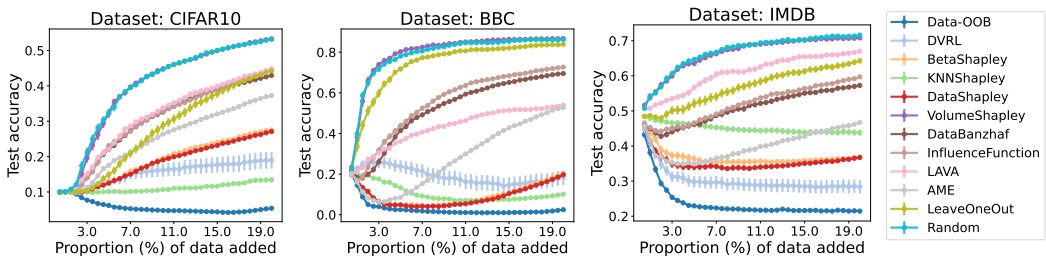

Figure 10: **Point addition experiment on non-tabular datasets.** Test accuracy curves as a function of the least valuable data points added. Lower curve indicates better data valuation algorithm. The error bar indicates a 95% confidence interval based on 50 independent experiments. Data-OOB exhibits superior performance.

is named to prevent collision with the PyTorch DataLoader, prepares the dataset into the training, testing, and validation datasets for experiments.

**Register**   The Register object allows us to assign a unique string identifier to each dataset. For instance, OpenDataVal provides various image datasets ('mnist10', 'cifar10', 'cifar100', 'cifar10-embeddings'), text datasets ('bbc-embeddings', 'imdb-embeddings'), and tabular datasets ('iris', 'digits'. 'breast_cancer', '2dplanes', 'electricity', 'MiniBooNE', 'pol', 'fried', 'nomao', and 'cred-itcard', just name a few). We will keep updating a list of available registered datasets. With the registered object, we can easily load the raw dataset, which allows us to have reproducible prepro-cessing. Also, as Code snippet 3 demonstrates, users can register and preprocess their own datasets. All the registered datasets are in the 'opendataval/dataloader/datasets' directory.

We highlight that two options are available to access the image and text datasets. The first option is the raw data using a PyTorch data loader [26], which allows direct handling and processing of the data. Alternatively, we also offer pretrained ResNet50 embeddings for image datasets [10] and pretrained DistilBERT embeddings for text datasets [30]. These embeddings provide a compact representation of the data, enabling efficient analysis and experimentation.

**DataFetcher**   Once a dataset is registered, it becomes accessible through the DataFetcher. The Code Snippet 4 demonstrates how to retrieve a registered dataset using a string identifier and split it into training, validation, and test datasets.

Additionally, we offer an alternative approach to retrieve datasets without the need for registration, as depicted in Code Snippet 5. For short-lived tasks, DataFetcher.from_data allows you to add data from numpy arrays.

To add synthetic noise, we can use one of the functions from the 'noisify' file in Code snippet 6. With this API, we can specify how to prepare datasets where the preprocessing needs to be consistent across experiments.

```
from opendataval.dataloader import Register
# Register a dataset
@Register(name="custom_dataset", one_hot=True, cacheable=True)
def read_custom_dataset():
    # read a raw dataset
    data = pd.read_csv('data_path.csv')

    # preprocess covariates and labels.
    covariates, labels = data.loc[:,:-1], data.loc[:,-1]
    return covariates, labels
```

Code snippet 3: The Register feature simplifies the process of reading and preprocessing raw datasets. The following code snippet demonstrates how to read a custom CSV dataset from my working directory and preprocess it to have covariates and labels. The function component can be easily modified. Additionally, users have the flexibility to download datasets from the internet and apply various normalization techniques as needed.

```
from opendataval.dataloader import DataFetcher

# Access a registered dataset
fetcher = DataFetcher(dataset_name="custom_dataset")

# Splitting and loading the data
fetcher.split_dataset_by_count(300, 100, 100)
x_train, y_train, x_valid, y_valid, x_test, y_test = fetcher.datapoints
```

Code snippet 4: With DataFetcher, users can conveniently access registered datasets and randomly split them into training, validation, and test sets, thereby facilitating smooth and reliable execution of subsequent experiments.

### D.2 Prediction Models

Data values are often computed in the context of how well they improve the prediction model performance. As such, many data valuation algorithms require a class of prediction models. In OpenDataVal, one can take a prediction model from a comprehensive collection of PyTorch classifiers and regressors, empowering users with a wide range of machine learning models for their data valuation tasks. Additionally, the framework provides convenient wrappers for scikit-learn models, enabling seamless integration of scikit-learn models into the OpenDataVal ecosystem. These wrappers ensure that scikit-learn models can be readily utilized and leveraged within the framework's data valuation pipeline.

To ease of use, OpenDataVal provides a ModelFactory that will load a prediction model from a string identifier of its name. The Code snippet 7 shows how to select and initialize a model.

### D.3 Data valuation algorithms

OpenDataVal provides eleven state-of-the-art data valuation algorithms in Table 1 with a unified API called DataEvaluator. All the data valuation algorithms are importable in the 'opendataval/dataval/' directory. One can develop a new data valuation algorithm by simply inheriting from the provided abstract DataEvaluator base class. DataEvaluator often requires a performance metric and a prediction model to be specified in order to determine data values. However, KNNShapley only produces data values for a KNN classifier. The injected prediction model will not be used to compute the data values but will be retained for evaluation tasks.

```
from opendataval.dataloader import DataFetcher

# Quick random regression task
covariates, labels = np.random.rand(500,5), np.random.rand(500,2)
fetcher = DataFetcher.from_data(covariates, labels, one_hot=False)

# Splitting and loading the data
fetcher.split_dataset_by_indices([i for i in range(300)],
                                 [i for i in range(300,400)],
                                 [i for i in range(400,500)])
x_train, y_train, x_valid, y_valid, x_test, y_test = fetcher.datapoints
```

Code snippet 5: Even if a dataset is not registered, it can be readily loaded. This code snippet demonstrates how to create a DataFetcher object using 'numpy' arrays for covariates and labels allowing seamless access to the dataset. Also, this code snippet shows how to split the dataset by preset indicies.

```
from opendataval.dataloader.noisify import add_gauss_noise, mix_labels
# Split data
fetcher.split_dataset_by_count(300, 100, 100)

# Adds noise
fetcher.noisify(mix_labels, noise_rate=0.2)
fetcher.noisify(add_gauss_noise, noise_rate=0.2, mu=0, sigma=1)
```

Code snippet 6: In our benchmarking analysis, we utilized a synthetically generated noisy dataset to assess the quality of data values. By employing the 'noisify' function, one can introduce label noise or feature noise.

```
from opendataval.model import ModelFactory

# Example 1: Loading PyTorch-based Logistic regression model
logistic_model = ModelFactory("logisticregression", fetcher, device="cuda")

# Example 2: Loading scikit-learn model
linear_model = ModelFactory("sklinreg", fetcher, fit_intercept=True)
```

Code snippet 7: Two examples of loading prediction models from PyTorch and scikit-learn using a string identifier. Available string identifiers include 'logisticregression', 'classifiermlp', 'regressionmlp', 'lenet', and 'sklogreg', to name a few. A full list of identifiers is defined in the 'opendataval/model/__init__.py' file.

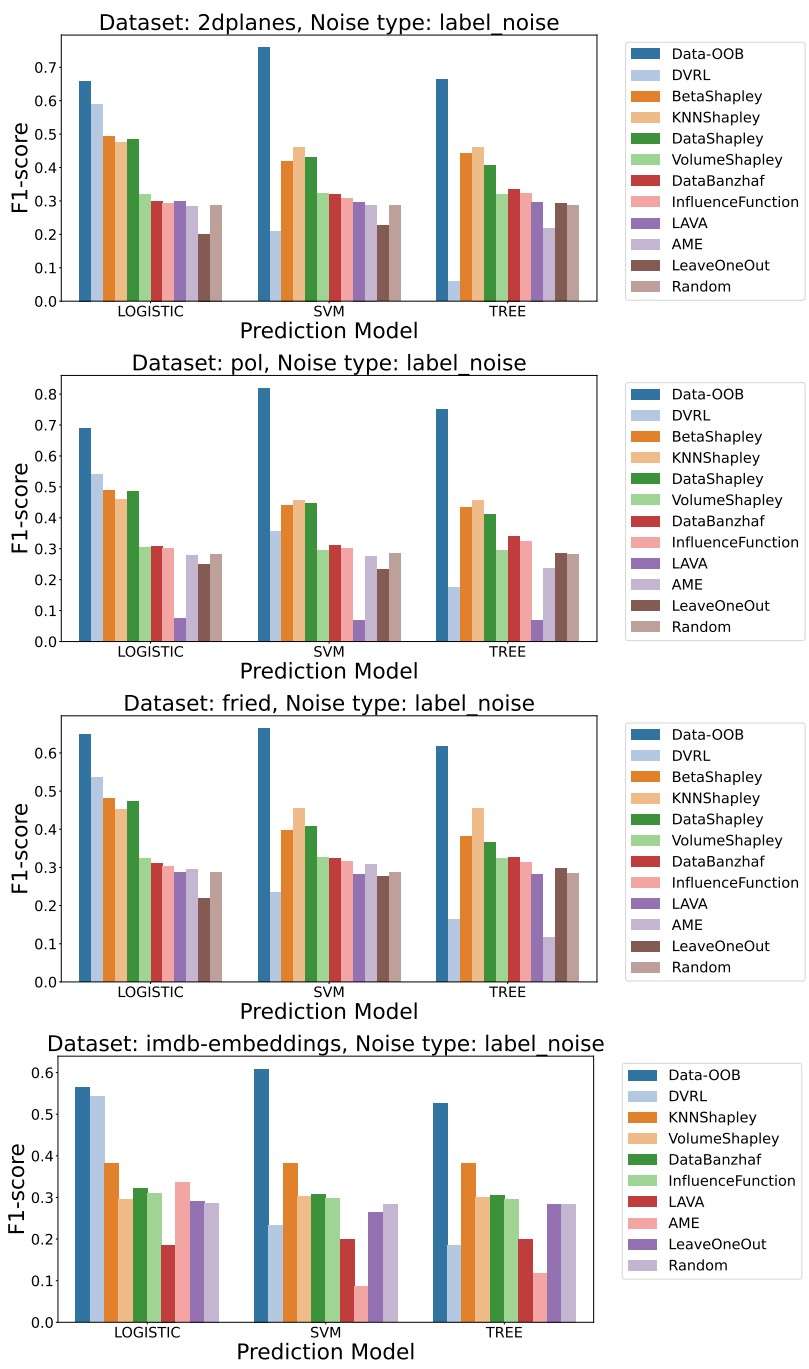

Figure 11: **Noisy label data detection across different prediction models.** F1-score of different data valuation algorithms on the four datasets. The higher the F1-score is, the better the data valuation algorithm is. Across different prediction models, Data-OOB generally outperforms other data valuation algorithms.

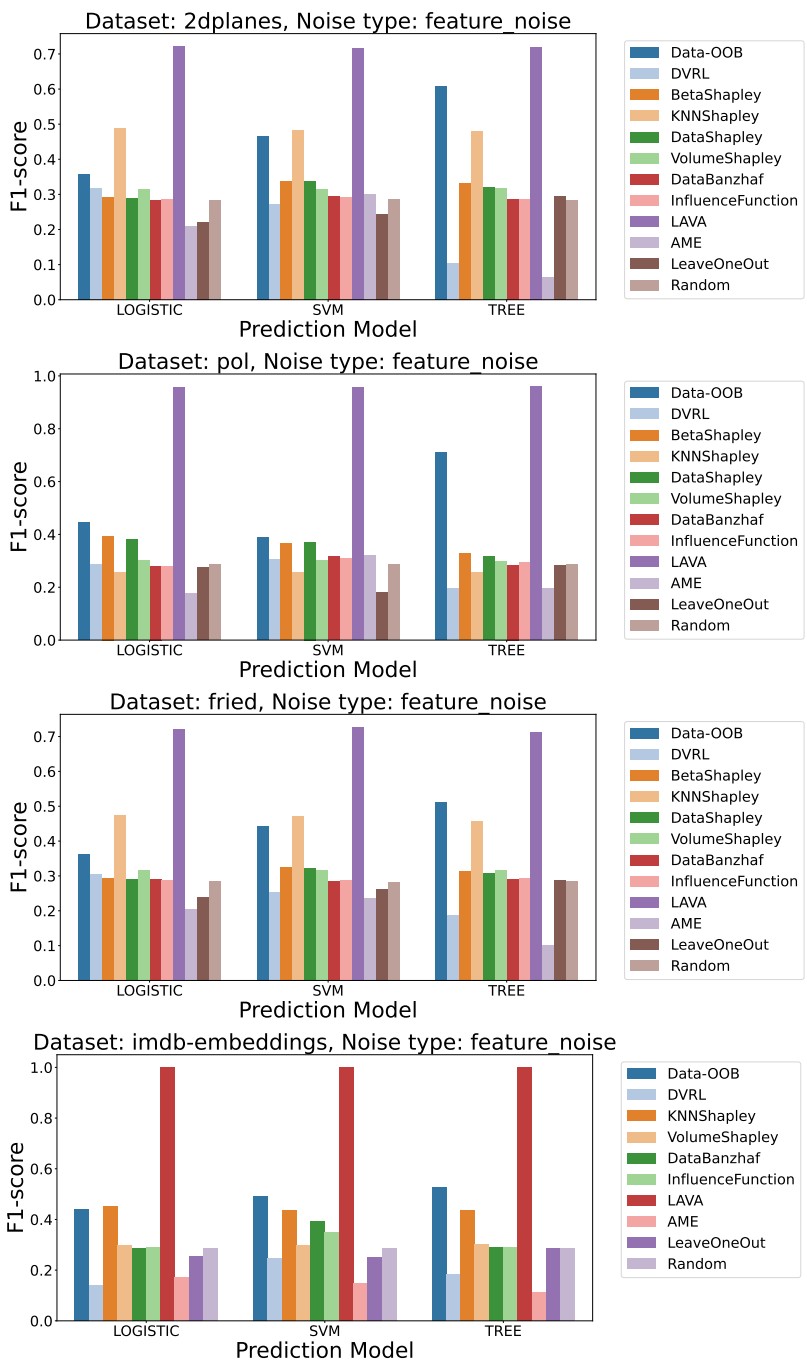

Figure 12: **Noisy feature data detection across different prediction models.** F1-score of different data valuation algorithms on the four datasets. The higher the F1-score is, the better the data valuation algorithm is. LAVA significantly outperforms other data valuation algorithms.

