# OpenReview forum: "OpenDataVal: a Unified Benchmark for Data Valuation"
_NeurIPS.cc/2023/Track/Datasets_and_Benchmarks — NeurIPS 2023 Datasets and Benchmarks Poster_

### Official Review · Reviewer_2UEH · 2023-07-19
**Important contribution that will benefit the data valuation research community**

**Rating:** 9
**Confidence:** 5

**Strengths:**

In my view, this is overall a very strong submission and is a very good fit for the dataset and benchmarks track. The focus on open and reproducible benchmarking is likely to greatly benefit members of the ML research community interested in data valuation (for a variety of applications) and may attract additional interest in the topic area (i.e. by making it easier to get started with existing implemenations and by making it easier to evaluate new approaches).

Significance of contribution: high. While no new algorithms or techniques are proposed, this is an area where benchmarking, aggregating implementations, and standardizing practice can be extremely useful. I believe this paper/resource could become quite central in providing some collective scaffolding for data valuation research going forward.

Relevance to broader research community: Data valuation is likely to be relevant to a good portion of the ML community. There is a great interest in interdisciplinary work using these approaches as well in economic (e.g. data markets) and sociology (e.g. data-related collective action). While there's not much discussion of applied uses here, I think the collective scaffolding benefits described above will flow downstream to "applied data valuation".

Quality: implementation and writing are all high quality (more comments on this below).

Ethical and social implications: While the paper does not have much room allocated for explicitly discussing ethical/societal implications, I didn't see any major risks or concerns with the work.

It could be worth a brief line of discussion on how making data valuation more accessible might impact the data economy (e.g. could data values be used to exclude people from markets or other systems), but ultimately I think these questions will be part of a long line of future work. I think overall, making benchmarking better falls firmly into the camp of benefits (transparency, openness, replicability, etc.) outweighing speculative harms.

**Additional Feedback:**

Thanks to the authors for this contribution to the research community!

**Clarity:**

The paper is very clear throughout, including the motivation of the work, the brief description and taxonomy of existing methods, and the explanation of benchmarking experiments that were conducted and the results.

**Correctness:**

The choice of data valuation algorithms seems reasonable and comprehensive.

The code is nicely documented and organized, such that readers can take a look at the implementation of the method they're interested in, find the relevant paper, quickly understand key parameters like cardinality, etc.

Dataset choices are reasonable albeit a bit arbitrary (and *some* choice has to be made).

Overall, I didn't identify any major issues with correctness of the paper. I think this work will serve to increase the rigor of data valuation research.

**Documentation:**

The paper and code provide helpful documentation regarding key details, source, etc. The code is hosted on GitHub. As I understand, the benchmarking framework is usable today (users can submit PRs to participate in the shared leaderboard, or download locally to try out their own new method in private).

**Ethics:**

As noted above, no major concerns here.

**Limitations:**

I didn't identify any major societal concerns of methods limitations that were missing here. As noted above, the discussion of societal impact is on the shorter side, but this is fair given the nature of the contribution.

**Opportunities For Improvement:**

As noted above, I found this to be an extremely strong paper overall. The following opportunities for improvements all rank as fairly minor:
- Not much room for discussion, but this is reasonable given the intended contribution of the paper.
- The datasets felt somewhat arbitrary (I wasn't sure if I missed additional justification details in an appendix), but this is very reasonable given swapping in one's own dataset of interest should be very easy. And to be fair, the chosen 9 do cover different sample sizes and characteristics.
- It might be interesting to in future to further characterize the different dataset choices, e.g. in terms of the underlying data-generating processes, the distribution of observations amongst contributors, etc. This could be a fruitful line of future work however.

**Relation To Prior Work:**

Good coverage of many data valuation techniques. I was a bit surprised to see Koh and Liang's influence function work not mentioned, as my understanding is that work (and its high profile at ICML) really kicked off a lot of the research interest in the topic in the ML community, which was previously of more interest to the statistics community (e.g. the key citations in the influence function paper could also be worth noting here -- it's both interesting and helps provide some insight into the theoretical framework these techniques emerge from).

However, I'm not sure how much the goal here is to emphasize the history of data valuation vs. describing currently popular techniques, so I leave it up to the authors if they want to add more "historical" literature.

**Summary And Contributions:**

This paper aims to provide standardization for comparing different data valuation techniques. Specifically, the paper presents and describes an open-source code package and leaderboard website. The authors also provide aggregated implementations of nine techniques with a qualitative overview of their strengths and a initial set of benchmark comparisons in noisy label detection, noisy feature detection, and add/remove experiments.

---

> ### Author Response · Authors · 2023-08-17
> **Response to Reviewer 2UEH**
>
> We thank the reviewer for appreciating the merits of our work with detailed comments.
>
> **Discussion on the impact of data valuation:** In the updated paper, we have added discussions on the potential impacts when data valuation is deployed in data marketplaces and federated learning scenarios.
>
> **Regarding the Koh and Liang paper:** We have considered the leave-one-out method as a proxy of Koh and Liang’s method since it fairly approximates the influence function in simple settings, which has also been observed in Section 4.1 of the original paper [1]. However, we agree with the reviewer that the paper opened many impactful directions in ML and deserves to be considered as relevant literature. We will add their algorithm in OpenDataVal soon.
>
> **Reference**
>
> [1] Koh, Pang Wei, and Percy Liang. "Understanding black-box predictions via influence functions." International conference on machine learning. PMLR, 2017.

---

> > ### Comment · Reviewer_2UEH · 2023-08-29
> > **Appreciate the response**
> >
> > Thanks to the authors for the response. I don't have too much more to add on this specific thread.

---

> > > ### Author Response · Authors · 2023-08-29
> > > **Thank you!**
> > >
> > > Thank the reviewer again for their constructive comments!

---

### Official Review · Reviewer_8KNo · 2023-07-21
**Although this paper is a valuable contribution for unifying different data valuation metrics, there is still room for improvement.**

**Rating:** 7
**Confidence:** 4
**Clarity:** The paper is well written and easy to…

**Strengths:**

This paper's stregths are as follows:

- The paper is well written and easy to follow.
- It provides the readers with a unified tool to implement, use and compare different data valuation metrics.
- Most of the well-known model-based metrics are included. This paper could be a good starting point for a student researcher to learn about these different metrics.
- Although the final insgihts (the fact that there is no "one fits all" type solution when it comes to data valuation metrics) are predictable and as observed in other works too, the downstream machine learning tasks used to draw these conclusions are sufficiently comprehensive for comparing the performance of different metrics.

**Additional Feedback:**

In addition to the weaknesses I listed above, I have the following additional comments:

- In the appendix, authors mention that in DataShapley, Monte Carlo algorithm is used to approximate the Shapley value, whereas for the case of DataBanzhaf, which is relatively similar to the DataShapley, estimates are not used. This selection is not clear to me.

- Page 4, line 95: the sentence describing DVRL is problematic.

- For the point removal and addition experiments, why did you only use the logistic regression model, which is a relatively simple task?

**Correctness:**

The claims made in this work seem correct. Authors benchmark some of the well-known data valuation techniques considering a well-established downstream machine learning tasks. There can always be more benchmarking tasks but the ones included in this work are treated carefully.

**Documentation:**

Yes, there is sufficient detail for reproducibility. The github repo is stated in the paper and authors describe how their framework is used for the experiments.

**Ethics:**

No.

**Limitations:**

- Authors address some current limitations and future directions in the conclusion section.
- Currently, there is no discussion on the potential negative societal impacts of their work. One negativity could be the fact that for data valuation techniques described in this paper, data owners should share their dataset, which would have security/privcacy implications.

**Opportunities For Improvement:**

Here are some of the weaknesses of the paper:

- Authors only consider model-based data valuation metrics but omit model-free valuation tools that study the intrinsic statistical properties of the data points. A truly "unified" data valuation framework should at least mention these approaches with a possibility to integrate them to the framework.

- In line with my previous comment, I would love to see how one can combine model-free and model-dependent data valuation metrics to achieve potentially superior performance in the downstream tasks described in the paper.

- In the data valuation environment, data owners can attack the system to increase their valuation through malicious behavior such as data duplication. Does the proposed framework cover such cases? Robustness against such attacks should be another criterion in selecting the valuation metric for the task at hand.

- The paper only considers point-by-point data valuation. What about data valuation in distributed/federated settings, where the data owners are not willing to share their data for privacy reasons? Can this framework be extended to the case in which data owners (datasets) are valued (through their submitted gradients)?

**Relation To Prior Work:**

The paper discusses its relation to existing literature on benchmarking/surveying data valuation algorithms.

**Summary And Contributions:**

In this work, authors propose a unified benchmarking framework, called OpenDataVal, to implement and compare some of the well-known data valuation metrics. In doing so, users of this framework can determine the value of individual data points for comparison purposes as well as investigate the performance of different data valuation metrics considering various downstream tasks such as noisy data detection and point removal/addition. Authors also describe how a user can add their own data valuation metric to their framework and present a leaderboard of the benchmark users based on their performance in a pre-determined noise detedction task.

---

> ### Author Response · Authors · 2023-08-17
> **Response to Reviewer 8KNo**
>
> We thank the reviewer for providing invaluable and overall positive comments.
>
> **Regarding model-free data valuation algorithm:** We have included a model-free data valuation algorithm called LAVA [1] in OpenDataVal. We find that it performs well in identifying data with perturbed features, though it is less powerful in detecting mislabeled data. Given these observations, we agree with the reviewer that a promising direction of research is to explore combining multiple data valuation algorithms where each captures different aspects of data values (e.g., Data-OOB for identifying noisy label data and LAVA for identifying noisy feature data).
>
> **Regarding the robustness and the societal impacts:** We greatly appreciate the reviewer for these thought-provoking comments. We agree with the reviewer that the robustness against attacks and data security/privacy issues will be important topics when data values are employed in real-world scenarios. In the updated paper, we have added discussions on the potential impacts when data valuation is deployed in data marketplaces and federated learning scenarios in the revision.
>
> **Regarding the DataBanZhaf:** Thank you for this question. As for the DataBanZhaf, we followed the original implementation, which does not directly estimate the marginal contribution. That is why do not use the marginal contribution estimates in computing DataBanZhaf.
>
> **Regarding the DVRL:** Thank you for pointing out this issue. We have clarified it in the revision.
>
> **Regarding the use of the logistic regression in point removal and addition experiments:** We mainly used the logistic model in the previous manuscript as it is one of the most fundamental and commonly used classification models. With OpenDataVal, users can easily change this model to any machine learning models. Related to this topic, please see the second item point of *Response to all reviewers* post. We have added other prediction models and the main results are consistent.
>
> **Reference**
>
> [1] Just, Hoang Anh, Feiyang Kang, Jiachen T. Wang, Yi Zeng, Myeongseob Ko, Ming Jin, and Ruoxi Jia. "Lava: Data valuation without pre-specified learning algorithms." International Conference on Learning Representation (2023).

---

> > ### Comment · Reviewer_8KNo · 2023-08-26
> >
> > Thank you to the authors for addressing my comments and suggestions in the revised manuscript. With the additional discussions and experiments, the manuscript has improved. With that, I will stick to my original rating of 7.

---

> > > ### Author Response · Authors · 2023-08-27
> > > **Thank you!**
> > >
> > > We are glad that we have addressed your concerns. Thank you again for thought-provoking comments.

---

### Official Review · Reviewer_CYt9 · 2023-07-21
**A comprehensive and off-the-shelf benchmark framework that enables comparison of data valuation algorithms over different perspectives**

**Rating:** 7
**Confidence:** 2

**Strengths:**

1. [significance of the contribution & relevance to the broader research community] This paper provides a solid benchmark for evaluating data valuation functions because of the diverse dataset inclusion, and evaluation metrics (performance on downstream tasks). It provides the performance insights of state-of-arts algorithms as well as an easy-to-use API to evaluate newly developed algorithms. I believe this benchmark framework and the insights obtained can provide convenience to the broader research community for data diagnosing and data quality improvement.
2. [quality of research] This paper employs solid methodologies in benchmark establishment, experiments. Most design choices such as dataset selection, start-of-arts algorithms selections, downstream tasks are well justified.


**Additional Feedback:**

Typo: Page 3 "can seen" -> "can be seen"

**Clarity:**

The paper is very well-written, with a clear background introduction, literature review, and organization, which is friendly to readers not familiar with this field.

**Correctness:**

The evaluation methods and experiment design are appropriate and performed correctly.

**Documentation:**

Yes

**Limitations:**

Data valuation algorithms may demonstrate different performance when the model used to compute the utility varies. This paper only uses the logistic regression as the underlying model. The paper adds some discussion or experiments about how different underlying model may impact the performance of different data valuation function.

**Opportunities For Improvement:**

See limitations.

**Relation To Prior Work:**

Yes.

**Summary And Contributions:**

This paper establishes a benchmark framework to evaluate data valuation algorithms. This framework has the following features:
1. It contains diverse datasets used to compare different algorithms.
2. It provides the evaluation metrics on different perspectives, including noisy label detection, noisy feature detection, point removal, point addition.
3. It provides baseline performance of several state-of-art data valuation algorithms based on various main methodologies.
4. The framework has easy-to-use API.

Based on the baseline performance of the state-of-arts algorithms, the paper reveals no algorithm dominate the others on all four perspectives.

---

> ### Author Response · Authors · 2023-08-17
> **Response to Reviewer CYt9**
>
> We greatly thank the reviewer for appreciating our paper and providing constructive comments.
>
> **Regarding the robustness of our empirical findings:** Please see the second item point of *Response to all reviewers* post. We have added different prediction models in our evaluations. The main findings are consistent across prediction models.
>
> **Editorial comments:** Thank you for pointing out the typo. We have corrected it in the revised manuscript.

---

### Official Review · Reviewer_hFxZ · 2023-07-23

**Rating:** 5
**Confidence:** 3
**Clarity:** Yes.

**Strengths:**

The significance of the work lies in developing a benchmark for a trendy problem of data valuation, facilitating the comparison of different data valuation algorithms.

In addition, the paper is easy to follow and well-structured.

**Additional Feedback:**

None.

**Correctness:**

The evaluation part could benefit from a more in-depth analysis of the sensitivity of each method to their hyperparameter.

**Documentation:**

Yes

**Limitations:**

The authors have partially addressed the limitations.

**Opportunities For Improvement:**

The benchmark is a bit limited in scope. In particular, the data valuation algorithms considered assume a fixed validation set and a fixed prediction model. Existing papers have also proposed validation-free [1], and model-agnostic [2] data valuation algorithms, which are not included in the benchmark.

The codebase does not provide an easy way to tune the hyperparameters of each data valuation algorithm.

Because the hyperparameters are not exposed, it is not clear whether the presented results are artifacts of specific choices of hyperparameters or something systematic.

[1] Xu, Xinyi, Zhaoxuan Wu, Chuan Sheng Foo, and Bryan Kian Hsiang Low. "Validation-free and replication robust volume-based data valuation." Advances in Neural Information Processing Systems 34 (2021): 10837-10848.

[2] Just, Hoang Anh, Feiyang Kang, Jiachen T. Wang, Yi Zeng, Myeongseob Ko, Ming Jin, and Ruoxi Jia. "Lava: Data valuation without pre-specified learning algorithms." International Conference on Learning Representation (2023).


**Relation To Prior Work:**

The work has discussed some of the related work but could benefit from comparing it with the following benchmark: https://uvanlp.org/valda/



**Summary And Contributions:**

The paper presents a data valuation benchmark. The contributions include the integration of diverse datasets, diverse existing data valuation algorithms, and various evaluation methods.

---

> ### Author Response · Authors · 2023-08-17
> **Response to Reviewer hFxZ**
>
> We thank the reviewer for your time and helpful feedback.
>
> **Regarding validation-free and model-agnostic data valuation algorithms:** We greatly appreciate the reviewer's constructive suggestion. Please see the first item point of *Response to all reviewers* post. We have added the two methods you mentioned to our benchmark evaluations.
>
> **Regarding the hyperparameter tuning:** Thank you for this suggestion. Our updated codebase provides an easy way to tune hyperparameters of data valuation algorithms. To better guide the reviewer and users through this process, we have provided many Jupyter Notebook examples in [our GitHub](https://github.com/opendataval/opendataval/tree/main/examples). Each example includes easy-to-follow instructions, demonstrating how to change a prediction model or its hyperparameter. For instance, [Step 1-2] of [this Jupyter Notebook](https://github.com/opendataval/opendataval/blob/main/examples/custom-dataset_classification.ipynb) introduces how to apply different prediction models, and [Step 1-2] of [this Jupyter Notebook](https://github.com/opendataval/opendataval/blob/main/examples/pol_classification.ipynb) shows how to specify model hyperparameters. Our Jupyter Notebook examples will guide users to tune hyperparameters, eventually applying various data valuation algorithms in their own data analysis.
>
> **Regarding the robustness of our empirical findings:** Please see the second item point of *Response to all reviewers* post. We have added results for different prediction models.
>
> **Related work:** Thank you for sharing [Valda](https://uvanlp.org/valda/) with us. We have discussed Valda in the revision.

---

> > ### Author Response · Authors · 2023-08-26
> > **We would like to hear back from Reviewer hFxZ**
> >
> > Dear Reviewer hFxZ,
> >
> > We would like to follow up to see if our response and revised manuscript address your concerns or if you have any further questions. We would greatly appreciate the opportunity to discuss this further if our response has not already addressed your concerns. Thank you again!
> >
> > Best regards,
> > Authors

---

### Author Response · Authors · 2023-08-17
**Response to all reviewers**

We thank all the reviewers for their time and constructive feedback. In the revised paper that we have uploaded, we have carefully incorporated the reviewers’ helpful suggestions, and changes have been colored in red. Also, we respond to each of the reviewer’s comments in separate posts, and the main additions are summarized as follows.

- We have **added the two data valuation algorithms** [1,2] to OpenDataVal, and they have been included as baseline methods in benchmarking analysis (Section 4). As a brief summary, we find that the volume-based Shapley [1] is not effective compared to many other data valuation methods in various tasks. This could be due to its sole dependence on feature information, which has also been observed in the previous work [2]. Meanwhile, LAVA [2] performs well in noisy feature data detection, outperforming other methods. Interestingly, it is less effective in detecting mislabeled data, and this limitation potentially results from the choice of the transport cost function, one of the key hyperparameters in LAVA.
- We have also **added new experimental results** to demonstrate the robustness of our empirical findings across different prediction models. We conducted the noisy label/feature data detection experiments in the same settings as used in Section 4, but we considered two additional prediction models: a decision tree model or a support vector machine. As a summary, we find that our findings are fairly robust and consistently observed: Data-OOB outperforms other methods in noisy label data detection and LAVA is shown to be effective in noisy feature data detection. We have added this discussion in Appendix C.3.
- We have **added discussions** on the potential impacts when data valuation is deployed in data marketplaces and federated learning scenarios.

Once again, we thank all the reviewers for their encouraging and valuable comments.

**References**

[1] Xu, Xinyi, Zhaoxuan Wu, Chuan Sheng Foo, and Bryan Kian Hsiang Low. "Validation-free and replication robust volume-based data valuation." Advances in Neural Information Processing Systems 34 (2021): 10837-10848.

[2] Just, Hoang Anh, Feiyang Kang, Jiachen T. Wang, Yi Zeng, Myeongseob Ko, Ming Jin, and Ruoxi Jia. "Lava: Data valuation without pre-specified learning algorithms." International Conference on Learning Representation (2023).

---

### Decision · Program_Chairs · 2023-09-22

**Decision:**

Accept (Poster)

**Comment:**

The main weaknesses highlighted by reviewers have been addressed by authors in the updated version of this work, notably, (1) model-based vs model-free data valuation metrics and (2) enabling hyper-parameter tuning in the codebase. One small comment is that Reviewer 2UEH notes the arbitrary selection of the datasets and the authors do not address this comment in their responses -- authors should consider adding more details as to the justification of the inclusion of specific datasets in an appendix.

Thank you to all reviewers and authors for their timely and thoughtful engagement. Overall, this paper is clearly well above the acceptance threshold.